# The Effect of Sexual Intercourse during Pregnancy on Preterm Birth: Prospective Single-Center Cohort Study in Japan

**DOI:** 10.3390/healthcare11111657

**Published:** 2023-06-05

**Authors:** Yoshie Yo, Kaoru Kawasaki, Kaori Moriuchi, Reona Shiro, Masao Shimaoka, Noriomi Matsumura

**Affiliations:** 1Department of Obstetrics and Gynecology, Faculty of Medicine, Kindai University, 377-2 Ohnohigashi, Osaka-Sayama 589-8511, Osaka, Japan; 2Department of Obstetrics and Gynecology, Saiseikai Tondabayashi Hospital, 1-3-36 Koyodai, Tondabayashi 584-0082, Osaka, Japan

**Keywords:** sexual intercourse during pregnancy, bacterial vaginosis, preterm delivery

## Abstract

Several studies in Europe and the United States have shown that sexual intercourse (SI) during pregnancy is not associated with preterm birth. However, it is unclear whether these findings apply to pregnant Japanese women. The aim of this prospective cohort study was to elucidate the influence of SI during pregnancy on preterm birth in Japan. A total of 182 women who underwent antenatal care and delivery were included in this study. The frequency of SI was assessed using a questionnaire, and its association with preterm birth was analyzed. The results showed that SI during pregnancy was associated with a significantly higher cumulative preterm birth rate (*p* = 0.018), which was more pronounced for SI more than once a week (*p* < 0.0001). Multivariate analysis showed that SI, bacterial vaginosis in the second trimester, previous preterm birth, and smoking during pregnancy were independent risk factors for preterm birth. The combination of SI and second trimester bacterial vaginosis was associated with a 60% preterm birth rate, whereas either factor alone was associated with a lower rate, suggesting a synergistic effect (*p* < 0.0001). Future studies are needed to investigate the effect of prohibiting SI in pregnant women with bacterial vaginosis on preterm birth.

## 1. Introduction

Preterm birth is the leading cause of child mortality worldwide, accounting for 35% of neonatal deaths and 18% of deaths in children under 5 years old [1]. Even when infants survive, they face neurodevelopmental disorders, such as cerebral palsy, neurocognitive disorders, behavioral and motor impairments, and blindness [2]. Ascending infection from the vagina through the cervix to the uterus is a major cause of preterm birth [3]. Infection in the cervix and the chorioamniotic membrane activates leukocytes, resulting in increased inflammatory cytokines, including interleukin (IL)-6, IL-8, and IL-1β, as well as tumor necrosis factor-α. Collagen fibrils are degraded by extracellular matrix metalloproteinase-8 and -9, and proteolytic enzymes such as elastase, leading to cervical ripening and membrane rupture, as well as uterine contractions. Prostaglandins E2 and F2α promote uterine contractions and labor progress [4,5].

*Lactobacillus* species (spp.) metabolize glycogen and produce lactate acid in vaginal mucosal cells. As a result, the vaginal environment becomes acidic at pH 3.5–4.0, inhibiting bacterial growth other than those of *Lactobacillus* spp. Thus, *Lactobacillus* spp. become dominant during pregnancy due to a marked increase in circulating estrogen, preventing ascending genital tract infection [6,7]. Bacterial vaginosis is characterized by the depletion of the normal *Lactobacillus* spp.-dominant microbiota and an overgrowth of commensal anaerobic bacteria [8], and carries increased risks of chorioamnionitis, premature rupture of the membrane, and preterm birth [9]. African Americans are at higher risk for preterm birth, which is associated with approximately one-quarter of pregnant women contracting bacterial vaginosis [10]. Antibiotics did not prevent preterm birth in pregnant women with bacterial vaginosis in randomized controlled studies [11,12,13,14].

Sexual intercourse (SI) during pregnancy has been considered a possible cause of preterm birth due to bacterial overgrowth in the birth canal, increased serum oxytocin levels, and uterine contractions induced by physical stimulation [15,16,17]. On the other hand, several studies in Israel, the United States, and Finland have shown that SI was not associated with preterm birth [18,19,20]. Therefore, the National Institute of Health and Clinical Excellence (NICE) guidelines state that pregnant women should be informed that SI during pregnancy is not related to adverse pregnancy outcomes, including preterm birth [21]. However, previous studies which concluded that sexual intercourse was not associated with preterm birth were not randomized controlled studies. Sexually active subjects may be younger and have a lower risk of preterm birth. This study bias could affect the results. Therefore, it is not clear that sexual intercourse does not increase the preterm birth risk. Furthermore, these studies were conducted on Caucasians. The association between SI and preterm birth risk has not been investigated in Japan. SI frequency varies widely by country, race, and culture, and the Japanese are the least sexually active in the world [22]. Additionally, Caucasians have a *Lactobacillus* spp.-dominant vaginal microbiome and are less prone to bacterial vaginosis [23], but similar data are lacking for Japanese women. It is unclear whether the Caucasian data can be applied to the comparatively sexually inactive Japanese women.

This is the first prospective cohort study to investigate the influence of SI on preterm birth in Japan, the least sexually active country. The results of this study will provide evidence for whether lifestyle advice about SI during pregnancy should vary by culture, race, and country.

## 2. Materials and Methods

### 2.1. The Patients

A single-center prospective cohort study at Kindai University Hospital was conducted on patients who underwent antenatal care and delivery from May 2018 to September 2019. This study was approved by the Ethics Review Board of Kindai University (Approval No.: 29-212) and written consent was obtained from all subjects. All methods were performed in accordance with the relevant guidelines and regulations.

### 2.2. Questionnaire Survey

The sexual intercourse (SI) frequency in the second and in the third trimester were surveyed at 30 weeks of gestation and after delivery on day 1–4 postpartum, respectively. Subjects completed the questionnaire alone in a room to protect their privacy. Answer sheets were collected in a box out of sight of health care providers.

### 2.3. Bacterial Vaginosis Screening

The screening was performed in the first trimester (10–12 weeks of gestation) and second (24–27 weeks of gestation). Our recent analysis of 1495 pregnant Japanese women demonstrated that a Nugent score (NS) [24] of 7 or higher in the first trimester and a NS of 4 or higher in the second trimester were associated with preterm birth [25]. Based on these data, we defined bacterial vaginosis as an NS of 7 or higher in the first trimester and an NS of 4 or higher in the second trimester.

### 2.4. Data Extraction

Maternal age, parity, history of preterm birth, smoking during pregnancy, and pregnancy outcomes were extracted from the medical records.

### 2.5. Systematic Review of Worldwide SI Frequency during Pregnancy

A thorough search was conducted on PubMed/MEDLINE encompassing the years from 1981 to 2022 using the terms: “sexual intercourse”, “coitus”, “sexual behavior” and “pregnancy”, and 1401 articles were extracted. A total of 46 articles, including 32 articles reported from 1981 to 2016, were reviewed in a relevant systematic review [26], in addition to 14 articles reported thereafter, through October 2022.

### 2.6. Statistical Analysis

The primary endpoint was to analyze whether SI during pregnancy increases preterm birth. The assumption for calculating the number of subjects was that the preterm birth rate was predicted to be 20% in the sexually active group and 5% in the control group. Assuming a ratio of 1:1 for those 2 groups, a total of 172 cases were required in order to achieve a power of 80% for detecting a difference in proportion of 0.15 between the 2 groups at a 2-sided *p*-value of 0.05. Enrollment continued throughout the enrollment period to ensure the study’s accuracy. Cumulative preterm birth rates were calculated using the Kaplan–Meier method. Univariate analysis and multivariate analysis using the Cox proportional hazards model for preterm birth risk factors was performed with EZR version1.5.5 [27]. Other statistical analyses were performed using GraphPad Prism Version 7.0 (Graph Pad Software, San Diego, CA, USA). A *p* value < 0.05 was considered statistically significant.

## 3. Results

### 3.1. Sexual Intercourse during Pregnancy and Preterm Birth Rate

182 of the 205 eligible women completed the questionnaire on sexual intercourse (SI) during pregnancy. Patients at an increased risk of sexually transmitted diseases were not included. In total, 63 (34%) patients had SI at least once during pregnancy and 35 (19%) of those patients had SI after 28 weeks’ gestation. Pregnant women who had SI during pregnancy had a significantly higher cumulative preterm birth rate than those who did not (*p* = 0.018, Figure 1A). In addition, 13 (7%) patients who had SI at least once a week had a higher rate of preterm birth than the other groups (*p* < 0.0001, Figure 1B).

### 3.2. The Risk Factors for Preterm Birth

Only 96 subjects were screened for bacterial vaginosis in the first trimester because the remaining subjects were referred to our hospital in the second trimester or later. There was no significant difference in the preterm birth rate between pregnant women with bacterial vaginosis (NS ≥ 7) and those without bacterial vaginosis (NS ≤ 6) in the first trimester (Figure 2A). In the second trimester, however, pregnant women with bacterial vaginosis (NS ≥ 4) had a significantly higher preterm birth rate compared to those without bacterial vaginosis (NS ≤ 3) (*p* = 0.001, Figure 2B). In the first trimester, all (10/10) cases of bacterial vaginosis were treated with metronidazole vaginal tablets. In the second trimester, 52% (13/25) cases of bacterial vaginosis received metronidazole, and the preterm birth rates were 23% (3/13) in the treated group and 42% (5/12) in the non-treated group (*p* = 0.667). Previous history of preterm birth and smoking during pregnancy were significantly associated with an increased preterm birth rate (*p* < 0.0001 and *p* < 0.0001, respectively) (Figure 2C,D). Maternal age and parity were not significantly associated with the preterm birth rate (Figure 2E,F). Screening for chlamydial infections using a PCR test was performed in the first or second trimester of pregnancy. In total, 180 out of 182 subjects were tested, of whom 1 tested positive. The patient delivered at 40 weeks of gestation.

Cox proportional hazards regression analysis was performed in order to evaluate risk factors for preterm birth. In univariate analysis, SI, bacterial vaginosis in the second trimester, previous preterm birth, and smoking during pregnancy were significantly associated with preterm birth. They were independent risk factors in multivariate analysis (Table 1).

### 3.3. Sexual Intercourse, Bacterial Vaginosis, and Preterm Birth

The combination of SI and bacterial vaginosis in the second trimester resulted in a 60% preterm birth rate, whereas either factor alone resulted in a lower rate (log-rank test, *p* < 0.0001), indicating a synergistic effect (Figure 3).

No synergistic effect was observed in the combination of SI with previous preterm birth, smoking during pregnancy, or bacterial vaginosis in the first trimester (Appendix A).

Of the 54 sexually active patients, 26 patients used condoms and 28 patients did not. There was no significant difference in the cumulative preterm birth rate between these two groups (Appendix A). In cases with bacterial vaginosis in the second trimester, there was no difference in the cumulative preterm birth rate between patients who reported condom use and those who reported non-use (Appendix A).

### 3.4. Systematic Review on Sexual Intercourse during Pregnancy

The SI rate in the present cohort study was compared with studies in other countries. The SI rate during pregnancy was described in 13 articles in 9 countries from 1981 to 2022 (Table 2) [28,29,30,31,32,33,34,35,36,37,38,39].

The median (range) rate was 86 (97–35)%. The SI rate tended to be lower in East Asia, and the 34% observed in this study was the lowest. There were two trends in the change in the SI rate during pregnancy: four studies showed a decrease from the first to the second trimester [29,31,33,35], and four studies showed an increase from the first to the second trimester and a decrease in the third trimester [30,34,37,39]. In the present study, the rate was not investigated in the first trimester, but decreased from the first to the second trimester, as previously reported. Eight studies compared the SI rate before and after pregnancy (Table 3) [33,34,35,37,40,41,42,43]. The rate increased in 6.7 (0.7–16.5)% cases, decreased in 75.3 (37.4–99.3)%, and remained unchanged in 21.2 (0–46.1)%. In the present study, SI decreased in 85.7% of cases, increased in 0.5%, and remained unchanged in 6.6%.

## 4. Discussion

Sexual intercourse (SI) during pregnancy has not been associated with preterm birth in previous studies [18,19,20,44]. However, in the current cohort study, SI during pregnancy was associated with a higher risk of preterm birth (Figure 1A, Table 1). Only 7% of patients had SI more than once a week, in which case the risk of preterm birth significantly increased (Figure 1B). The primary endpoint was to analyze whether SI during pregnancy increased preterm birth, as mentioned. The *p* value < 0.05 shown in Figure 1 is meaningful because we were able to assess our hypothesis. However, we only demonstrated our hypothesis.

Bacterial vaginosis has been considered a preterm birth risk [45,46], and the same is true in pregnant Japanese women [25]. The present exploratory analysis combining SI during pregnancy and other risk factors for preterm birth (Figure 2, Table 1) [47,48,49] showed that the preterm birth rate significantly increased when SI and bacterial vaginosis were combined in the second trimester (Figure 3). The study in the United States found no association between SI frequency at 23–26 weeks of gestation and preterm birth, but pregnant women with frequent intercourse and bacterial vaginosis were at higher risk [17]. The results of this subgroup analysis are consistent with the present study. However, the present cohort had the lowest SI rate compared to previous studies in other countries (Table 2), which may be the reason for the difference in preterm birth risk from the other studies. The association between SI and preterm birth would not be shown in a cohort in which the majority of eligible women experience SI during pregnancy, when many of them do not have bacterial vaginosis.

Further investigation is required to elucidate how SI leads to preterm birth in pregnant women with bacterial vaginosis. One possible mechanism is that SI pushes vaginal bacteria into the cervix. Inflammatory cytokines and prostaglandins in the vaginal secretions of women with bacterial vaginosis are significantly higher than in healthy women [50], resulting in preterm birth when they are pushed into the cervix by SI [51]. Prostaglandin D2 receptor polymorphisms are associated with an increased risk of postcoital preterm birth [52]. Semen also contains prostaglandin E2, which induces cervical ripening and labor [53]. However, the present study found no difference in preterm birth rates between patients who reported condom use and those who reported non-use (Appendix A). Thus, the influence of semen on preterm birth may be limited.

The results of this study suggest that the treatment of bacterial vaginosis in pregnant women could be effective in preventing preterm birth. Our previous retrospective cohort study showed that an improvement in bacterial vaginosis with metronidazole tended to reduce preterm birth. However, in the overall population, metronidazole did not reduce preterm birth because of drug resistance [25]. Clinical trials in non-pregnant women have shown that SI can cause drug resistance in the treatment of bacterial vaginosis [54]. Therefore, if a pregnant woman is found to have bacterial vaginosis, it may be effective to treat the couple at the same time and advise against SI during pregnancy until treatment is completed. Such a clinical trial has not been conducted. Notably, bacterial vaginosis was not associated with preterm birth in the absence of SI (Figure 3). Lifestyle advice to pregnant women with bacterial vaginosis to avoid SI during pregnancy may be useful until a reliable treatment for bacterial vaginosis in pregnant women is established.

Further research is needed in order to include the strategies suggested by the present study in clinical practice. Firstly, the number of sexually active patients with bacterial vaginosis was small (n = 8, Figure 3) because a combined analysis of SI and bacterial vaginosis was not designed before the study began. Therefore, a larger study is required in order to conduct a statistically sound analysis of this combination as a primary outcome. Secondly, the number of patients tested for bacterial vaginosis in the first trimester was small because of the limited number of women who underwent antenatal care at our hospital in the first trimester. Future studies should perform the screening for bacterial vaginosis in the first trimester. Thirdly, feasibility studies on lifestyle advice prohibiting SI during pregnancy should be individualized by race and country because cultural background and personal values greatly influence SI. Japanese women are generally sexually inactive during non-pregnancy [22] and show reduced SI frequency during pregnancy due to concerns about adverse obstetric outcomes (Table 3). It may be easier to prohibit SI during pregnancy in Japan than in other countries.

## 5. Conclusions

The present study is the first to show the SI rate and frequency for pregnant Japanese women, which is extremely low compared to studies in other countries. This study also demonstrated that SI was associated with an increased risk of preterm birth. The exploratory analysis indicated the markedly increased risk of preterm birth when SI was combined with bacterial vaginosis, suggesting that treatment including lifestyle advice prohibiting SI for pregnant women with bacterial vaginosis may be beneficial. Further studies are necessary to investigate the reproducibility of the results in this study with larger sample sizes and to elucidate the effect of SI prohibition for pregnant women with bacterial vaginosis on preterm birth in randomized intervention trials.

## Figures and Tables

**Figure 1 healthcare-11-01657-f001:**
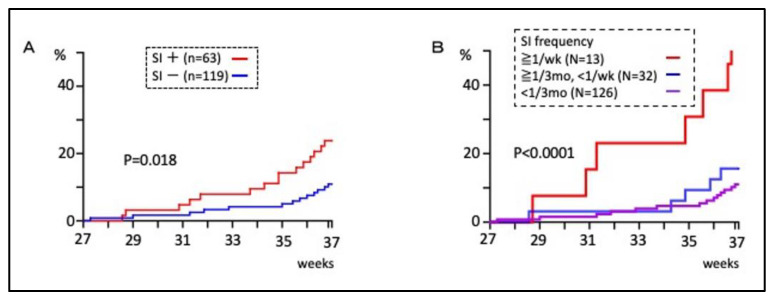
The association between sexual intercourse (SI) during pregnancy and the cumulative preterm birth rate. (**A**) Comparison based on whether or not there was SI, regardless of frequency. (**B**) Comparison among three groups divided by frequency of SI. The vertical and horizontal axes show the cumulative preterm birth rate (%) and the gestational age at preterm birth (weeks), respectively. wk—weeks; mo—months.

**Figure 2 healthcare-11-01657-f002:**
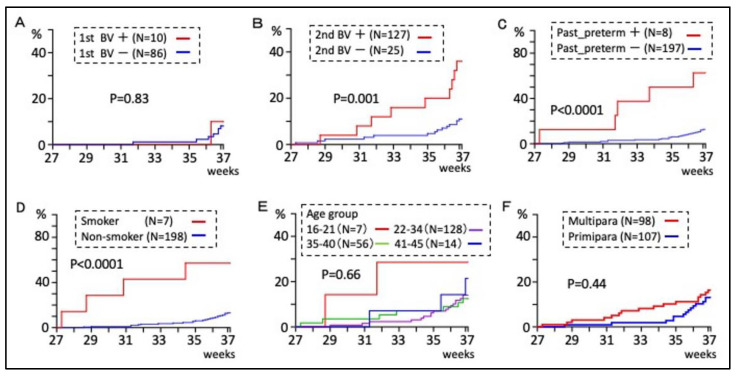
The association between the known risk factors for preterm birth and the cumulative preterm birth rate. (**A**) Comparison based on presence or absence of bacterial vaginosis (BV) in the first trimester. (**B**) Comparison based on presence or absence of BV in the second trimester. (**C**) Comparison based on presence or absence of a past history of preterm birth. (**D**) Comparison between smokers and non-smokers during pregnancy. (**E**) Comparison of four groups divided by maternal age. (**F**) Comparison between multipara and primipara.

**Figure 3 healthcare-11-01657-f003:**
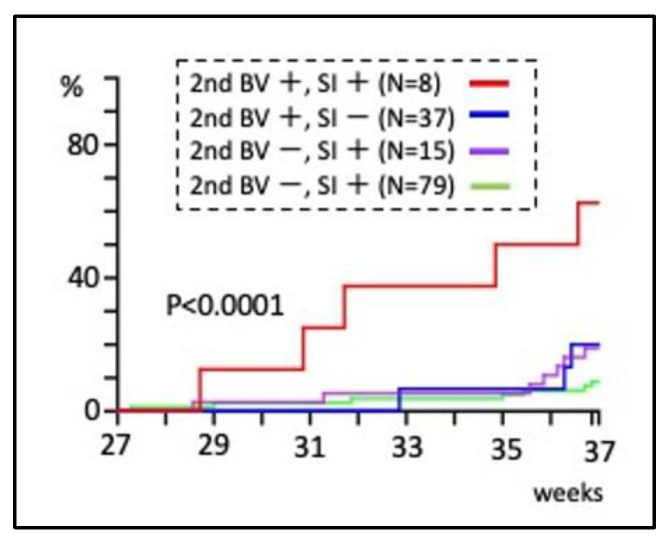
Combination of SI with BV in the second trimester and its association with the cumulative preterm birth rate.

**Table 1 healthcare-11-01657-t001:** Cox univariate and multivariate analysis of association between risk factors and preterm birth rate. PTB—preterm birth; BV—bacterial vaginosis; N/A—not applicable; SI—sexual intercourse.

	Univariate Analysis	*p* Value	Multivariate Analysis	*p* Value
Age	1.01 (0.94–1.08)	0.80	N/A	
Primipara	1.30 (0.63–2.66)	0.48	N/A	
Previous PTB	7.99 (3.04–20.99)	0.00002	6.02 (1.68–21.57)	0.006
Smoking	7.82 (2.72–22.49)	0.0001	4.54 (1.01–20.46)	0.049
BV in the 1st trimester	1.46 (0.18–12.15)	0.73	N/A	
BV in the 2nd trimester	4.00 (1.71–9.36)	0.001	3.53 (1.42–8.79)	0.007
SI during pregnancy	2.36 (1.12–4.97)	0.023	2.97 (1.24–7.16)	0.015

**Table 2 healthcare-11-01657-t002:** Systematic review on SI rate during pregnancy.

First Author	Reference	Year	Country	Region	Number	Overall (%)	1st Trimester (%)	2nd Trimester (%)	3rd Trimester (%)
Staruch M	[28]	2016	Poland	Eastern Europe	149	87	NA	NA	NA
Kulhawik R	[29]	2022			100	86	86	60	56
Yanikkerem E	[30]	2016	Turkey		298	84	79	84	65
Erene AS	[31]	2011			336	95	95	87	41
Kumar R	[32]	1981	England	Western Europe	119	90	90		65
Bartellas E	[33]	2000	Canada	North America	139	96	96	89	67
Iliyasu Z	[34]	2016	Nigeria	Western Africa	336	97	88	91	97
Naim M	[35]	2000	Pakistan	Southern Asia	150	89	89	82	74
Phan TC	[36]	2021	Vietnam	South-Eastern Asia	250	71	71		NA
Fok WY	[37]	2005	China	Eastern Asia	298	66	57	66	50
Chen L	[38]	2020			323	51	NA	NA	NA
Kong L	[39]	2019			406	51	12	49	15
Yo Y	This study		Japan		182	35	NA	31	19

**Table 3 healthcare-11-01657-t003:** Systematic review on the change in the SI rate from pre-pregnancy to pregnancy.

First Author	Reference	Year	Country	Number	Increased (%)	Decreased (%)	No change (%)	NA (%)
Adeyemi AB	[40]	2005	Nigelia	134	16.5	37.4	46.1	0
Iliyasu Z	[34]	2016	Nigelia	336	16.3	55.4	23.8	4.5
Guendler JA	[41]	2019	Brasil	262	7.6	64.9	27.5	0
Bartellas E	[33]	2000	Canada	139	5.8	71.2	23	0
Radoš SN	[42]	2014	Croatia	150	1.4	79.3	19.3	0
Branecka-Woźniak D	[43]	2020	Poland	181	10.5	89.5	0	0
Fok WY	[37]	2005	China	298	1.3	91.6	7	0
Naim M	[35]	2000	Pakistan	150	0.7	99.3	0	0
Yoshie Y	This study		Japan	182	0.5	85.7	6.6	7.1

## Data Availability

The data that support the findings of this study are available from the corresponding author.

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
