# Peer review of "The Effect of Sexual Intercourse during Pregnancy on Preterm Birth: Prospective Single-Center Cohort Study in Japan"

_healthcare, 2023, doi:10.3390/healthcare11111657_

Round 1
Reviewer 1 Report
The work is interesting because of the topicality of premature birth and possible risk factors, including sexual intercourse during pregnancy.
Introduction, methodology with results and discussion are clearly presented, and references contrast discussion and research. In this form, the work can be accepted as original because it is based on own research on chronic obstetric syndrome, premature birth.
Minor corrections refer to English medical expressions, i.e. proofreading, without special remarks.
Author Response
We really appreciate your efforts in reviewing our manuscript. We have corrected some medical terms.
Reviewer 2 Report
62. He added that sex life also depends on the age group and generation of young people. We observe a greater acceptance of young people regarding sexual behavior.
75. Did the study exclude patients at increased risk of sexually transmitted diseases, e.g. HIV?
88. Were pregnant patients tested for atypical infections - PCR test - Chlamydia, Mycoplasma, Ureoplasm?
I really like the comparison in the form of a table of data from different countries. Table 1 and 2.
The study lacks data on the sexual openness of the population and the conditions for excluding, for example, more than one partner and more risky behaviors from the period before and during pregnancy.
Author Response
Please see the revised version manuscript.
Reviewer 3 Report
Authors report their clinical experience concerning sexual intercorse (SI) in Japanese women during pregnancy and its relationship with preterm birth.
First, it is surprising that we should come to learn the definition of preterm birth only looking to the graphs...the graphs report a range between 27 to 37 weeks....Is this authors definition for preterm birth? r it should be form 23+1 weeks to 37+0 week's gestation?
Second point, authors should be aware that per se a p value < 0.05 does not demonstrate anything, especially when multiple variable may independently or dependently act. In this view, p value only inidcate that a clinical variable may be of some interest and should be confirmed by another investigation and clarification. Furthermore, as we are measuring a clinical effect on mothers, the calculation f relative risk (RR) or odds ratio (OR) are mandatory. Notwithstanding, authors can not conclude that "...bacterial vaginosis is a clinical risk factor for preterm birth" rather than calculating the multiple clinical variables using a stepwise logistic regression analysis!
Author Response
Dear Reviewer,
Reviewer 3
Authors report their clinical experience concerning sexual intercorse (SI) in Japanese women during pregnancy and its relationship with preterm birth.
Response: We wish to express our appreciation to your insightful comments on our paper. Our point-by-point response are detailed below.
First, it is surprising that we should come to learn the definition of preterm birth only looking to the graphs...the graphs report a range between 27 to 37 weeks....Is this authors definition for preterm birth? r it should be form 23+1 weeks to 37+0 week's gestation?
Response: Thank you for your comment. No subjects had given birth before 27 weeks of gestation. Therefore, the graphs were plotted from 27 to 37 weeks of gestation.
Second point, authors should be aware that per se a p value < 0.05 does not demonstrate anything, especially when multiple variable may independently or dependently act. In this view, p value only inidcate that a clinical variable may be of some interest and should be confirmed by another investigation and clarification.
Response: We appreciate your comment. The primary endpoint was to analyze whether SI during pregnancy increased preterm birth as mentioned in Line 113. The p value <0.05 shown in Figure 1 is meaningful because we were able to assess our hypothesis. However, as the reviewer pointed out, we only demonstrated our hypothesis. We have added this point in line 262-264.
Furthermore, as we are measuring a clinical effect on mothers, the calculation f relative risk (RR) or odds ratio (OR) are mandatory. Notwithstanding, authors cannot conclude that "...bacterial vaginosis is a clinical risk factor for preterm birth" rather than calculating the multiple clinical variables using a stepwise logistic regression analysis!
Response: Thank you for your comment. In the present study, the Cox proportional hazards model was used to assess the outcome, the time of delivery. Stepwise logistic regression analysis was inappropriate because it does not include time as a factor. Many studies have used Cox proportional hazards models to examine the association between preterm birth and its risk factors and treatment effects (Brubaker SG, et.al. Vaginal progesterone in women with twin gestations complicated by short cervix: a retrospective cohort study. BJOG. 2015;122:712-8. Tanz LJ et.al. Preterm Delivery and Maternal Cardiovascular Disease in Young and Middle-Aged Adult Women. Circulation. 2017;135:578-589. Waks AB, et.al. Developing a risk profile for spontaneous preterm birth and short interval to delivery among patients with threatened preterm labor. Am J Obstet Gynecol MFM. 2022;4:100727). Therefore, analyses using Cox proportional hazards models are appropriate in the present study.
For more details please see the revised version manuscript.